# Microglial Extracellular Vesicles as Modulators of Brain Microenvironment in Glioma

**DOI:** 10.3390/ijms232113165

**Published:** 2022-10-29

**Authors:** Myriam Catalano, Carmela Serpe, Cristina Limatola

**Affiliations:** 1Department of Physiology and Pharmacology, Sapienza University, 00185 Rome, Italy; 2Laboratory Affiliated to Istituto Pasteur Italia, Department of Physiology and Pharmacology, Sapienza University, 00185 Rome, Italy; 3IRCCS Neuromed, 86077 Pozzilli, Italy

**Keywords:** extracellular vesicles, microglia, glioma, CNS, immune evasion

## Abstract

Microglial cells represent the resident immune elements of the central nervous system, where they exert constant monitoring and contribute to preserving neuronal activity and function. In the context of glioblastoma (GBM), a common type of tumor originating in the brain, microglial cells deeply modify their phenotype, lose their homeostatic functions, invade the tumoral mass and support the growth and further invasion of the tumoral cells into the surrounding brain parenchyma. These modifications are, at least in part, induced by bidirectional communication among microglial and tumoral cells through the release of soluble molecules and extracellular vesicles (EVs). EVs produced by GBM and microglial cells transfer different kinds of biological information to receiving cells, deeply modifying their phenotype and activity and could represent important diagnostic markers and therapeutic targets. Recent evidence demonstrates that in GBM, microglial-derived EVs contribute to the immune suppression of the tumor microenvironment (TME), thus favoring GBM immune escape. In this review, we report the current knowledge on EV formation, biogenesis, cargo and functions, with a focus on the effects of microglia-derived EVs in GBM. What clearly emerges from this analysis is that we are at the beginning of a full understanding of the complete picture of the biological effects of microglial-derived EVs and that further investigations using multidisciplinary approaches are necessary to validate their use in GBM diagnosis and therapy.

## 1. Introduction

Primary brain tumors are the most malignant cancers in adults [1]. Among them, malignant glioma (glioblastoma multiforme, GBM) is the most frequent (with an incidence of 9.4/100,000), with an overall patient survival of 12–15 months from diagnosis [2]. Despite an aggressive therapeutical protocol that includes surgery, radio- and chemotherapy, tumor invasion into the surrounding brain parenchyma prevents its complete eradication and is the reason for recurrence.

Microglia constitute from 5 to 20% of all glial cell populations in the central nervous system (CNS) [3] and represent the resident immune cell population of the brain, continuously scanning surrounding cells and reacting to neuronal activity and to cellular alterations. Under inflammatory conditions, microglia display an active response that reduces CNS damage and supports neuronal tissue repair. Dysregulation or the over-activation of microglial immune response is a crucial cue in nearly all CNS diseases, including GBM [4,5].

In the GBM microenvironment, the communication between microglial and tumor cells is important for tumor growth as well as for the activation of microglia-mediated protective and tumor-promoting mechanisms. Microglial and GBM cells bidirectionally communicate through direct cell-to-cell contact or via soluble released factors such as chemokines, cytokines, growth factors and matrix metalloproteinase. In addition to these soluble factors, extracellular vesicles (EVs) represent a very well-established pathway of communication in the CNS [6].

In this review, we specifically discuss the contribution of EVs released by microglia on the growth and maintenance of GBM and on anti-tumor effects. An overview of microglia EV effects in glioma is provided in Figure 1.

Microglia may produce EVs with an anti-tumor function (in green) or protumor effect (in red). These EVs target microglia, astrocytes, and glioma cells. Microglial EVs exacerbate glioma by promoting the tumor-associated profile of microglia and increasing migration and invasion of tumor cells. Otherwise, microglial EVs contrast the tumor by reducing the pro-tumoral phenotype of microglia, reducing migration and invasion of glioma cells, enhancing the expression of glutamate transporters on astrocytes, and restoring homeostasis. The cargo of microglial EVs may depend on the microglial phenotype (neuroprotective or neurotoxic) and on their biogenesis process (see details in the text).

## 2. Extracellular Vesicles

All living cells are capable of secreting EVs. The production of EVs was initially described as a mechanism for the clearance of unnecessary compounds from the cells [7]. In 1967, researchers identified small lipid vesicles derived from whole serum as well as from platelets after ultracentrifugation. This material was initially referred to as “platelet dust” and later identified as composed of microparticles. In 1971, Aaroson described the membranous structures as EVs [8]. EVs of different sizes were visualized using electron microscopy and it took a long time before the hypothesis that considered EVs as experimental “artifacts” was finally excluded. In 1981, Trams observed that shed “microvesicles” collected from the culture medium of glioblastoma cell lines had a membrane composition similar to specific plasma membrane domains and that these vesicles induced specific effects on receiving cells [9]. Despite the initial evidence, for many years, EVs have been largely overlooked. Only one decade ago, the production of EVs started to be recognized as a new possible mechanism of intercellular communication [10]. EVs have heterogeneous structures delimited by a lipid bilayer and because they lack a functional nucleus, cannot self-replicate. EVs are divided into two categories based on dimension: medium/large EVs (m/lEVs) and small EVs (sEVs). These two populations differ in size, in the mechanism of their formation and in cargo. Concerning the cargo, all EVs can transport molecules with biological activity such as proteins, lipids, and nucleic acids, including DNA and different kinds of RNAs, such as mRNA and microRNA (miRNA). miRNAs are small non-coding molecules able to bind to complementary sequences in the 3′-untranslated regions (3′UTRs) of target mRNAs, defining the post-transcriptional regulation of different genes [11].

Much has been learned on the content of EVs but the identification of cargo specific for small or medium/large vesicles is an ongoing area of intense research. 

Because of their capacity to exchange molecules between cells, EVs act as signaling structures both under physiological and pathological conditions. Specifically, in pathological conditions, EVs might act in favor of disease, such as in multiple sclerosis [12], Alzheimer’s [13,14], prion [15], and Huntington’s disease [16]. These aspects, together with the identification of m/lEVs in different body fluids, increased the interest in research to elucidate the vesicle functions in different contexts [17]. In addition, EVs transport biologically active factors across the blood–brain barrier (BBB) and the choroid plexus [18], making them potential tools for the early diagnosis of neurological disease [19] and potential vehicles for non-invasive therapies [20]. 

### 2.1. Medium/Large EVs

According to the physical characterization and to the guidelines of the International Society of Extracellular Vesicles (ISEV) [21], m/lEVs comprise all the vesicles with a diameter larger than 200 nm. They are produced by the direct invagination of the plasma membrane and then released into the extracellular space [22]. 

Each cell type changes the EVs composition according to its physiological state, regardless of its size, with peculiar lipids, proteins, and nucleic acids content [23]. The process of plasma membrane blebbing is typical of m/lEVs, even if the underlying mechanisms remain incompletely understood [23]. 

Recently, it was reported that the process involved in membrane curvatures during EV formation requires the interaction of the arrestin domain-containing protein-1 (ARRDC1) with the endosomal tumor susceptibility protein 101 (TSG101) and that this interaction leads to the transfer of TSG101 from the endosomal compartment to the plasma membrane. The presence of TSG101 in the plasma membrane allows the release of m/lEVs containing TSG101 and ARRDC1, together with other molecules. Other mechanisms are involved in the process of plasma membrane curvature. For example, the enrichment of proteins at the cellular periphery and the pressure generated by their interaction could contribute to shaping changes and curvature. This process indicates that the enrichment of protein at the site of m/lEVs budding might be a stimulus sufficient to start vesicle formation [24]. Another important factor is the alteration in the lipid composition that modifies membrane rigidity and curvature [25], key events for m/lEV formation. Phospholipids are made up of hydrocarbon tails and large head groups that give them a conical shape, with an irregular distribution. The membrane curvature is determined by the distribution of phospholipids in the plasma membrane but also by the presence of aminophospholipid translocases, such as flippases and floppases, similarly to what is described for the Golgi vesicles [26]. 

### 2.2. Small EVs

Differently, sEVs are vesicles with a diameter smaller than 200 nm [21]. At the beginning of the 1980s, a pathway that includes endocytosis was described for sEV formation. It includes the internalization of extracellular ligands and cellular components that will then be transferred again to the plasma membrane and/or degraded [27]. Upon degradation, early endosomes will be the first to be produced, followed by late endosomes [28], which accumulate intraluminal vesicles (ILVs), or multivesicular endosomes (MVBs), inside. Proteins and lipids contained in the MVBs contribute to the curvature of the early endosomal membrane. Mostly, MVBs fuse with lysosomes and their content is degraded by hydrolases. In other cases, they can fuse with clusters of the plasma membrane enriched in tetraspanin CD63, lysosomal-associated membrane proteins (LAMP1 and LAMP2) and other molecules typical of late endosomes, releasing the content into the extracellular space [29,30]. 

To exert biological effects, EVs can either be internalized or activate receptor–ligand signaling on the surface of target cells. In the last few years, the mechanisms of EVs–cell interaction have also been investigated, taking advantage of drugs or antibodies to block specific signaling pathways. These studies revealed that EVs can be internalized by target cells through different mechanisms, such as clathrin-mediated endocytosis, phagocytosis, micropinocytosis and plasma or endosomal membrane fusion [31,32,33,34,35], in addition to protein–protein interactions mediated by tetraspanins (as CD63, CD9 and CD81), integrins and immunoglobulins, proteoglycan, and lectins [36]. 

## 3. Microglial EVs: Biogenesis, Markers and Functions

Microglial-derived EVs carry lipids, proteins, DNA and RNAs to target cells, conveying different packages of information [37] according to their activation state. One function of microglia-derived EVs is to transfer cytokines to distant brain regions, regulating inflammation [38].

Since no cell-specific mechanism of EV formation has been identified so far, microglial EV biogenesis does not differ from what is described for all the other cells (see Section 2). The number of vesicles released by microglial cells can, however, be regulated: ATP-mediated activation of P2X7 receptors is involved in EV release in N9 cells and in primary microglial cells. ATP stimulation also affects EVs’ protein content, modulating the level of proteins involved in cell adhesion and phagocytosis, energy metabolism and in autophagy [39]. Furthermore, ATP stimulation specifically increases Tau protein expression in microglial EVs [40]. Microglial-derived EVs are also involved in the process of synaptic pruning, tagging synapses through the complement protein C1q [41]. 

Another stimulus for EV production by microglial cells is the protein capsaicin: the activation of its receptor, the transient receptor potential vanilloid type I (TRPVI), increases the production of m/lEVs [42]. α-synuclein (α-syn) also stimulates the production of microglia-derived EVs leading to the formation of vesicles enriched in TNF-α and in the major histocompatibility complex (MCH)-II receptor. α-syn can be also released by microglial EVs [43]. 

Microglial-derived EVs play modulatory roles in pathological conditions. In traumatic brain injuries (TBI), microglial-derived EVs propagate inflammatory signals [44]. Microglial stimulation with LPS-enriched EVs with IL1-β, TNF-α, CCL2, IL-6, NOS-2 and miR-155, thus transferring the pro-inflammatory information to the recipient cells. The proteomic analysis of sEVs derived from rat microglia isolated from the spinal cord and the cerebral cortex of control and LPS-treated animals revealed different compositions and inflammatory effects [45]. All vesicles derived from microglia express the typical exosomal markers tetraspanin CD81, Anxa2, S100, and C1q. The sEVs derived from the microglia isolated from healthy animals are enriched in proteins involved in neurodevelopment such as axonal growth promoters, neuroprotective factors, and promoters of neurites outgrowth. Instead, sEVs derived from microglia isolated from LPS-treated animals transport proteins involved in chemotaxis, in the activation of the NFkB pathway, metabolic processes, and transport pro-inflammatory chemokines and cytokines such as IL-6 and TNF-α [45].

Other regulators of microglial EV release are neurotransmitters. For example, serotonin (5-HT) induces EV release from microglia through the stimulation of 5-HTR2 and 5-HTR4. The activation of 5-HTRs increases the intracellular levels of Ca^2+^ and cAMP, both enhancing EV secretion. Even if a direct observation is lacking, authors postulate a specific effect of microglia-derived EVs on serotoninergic neurons [46], widening the mechanisms of communication between neurons and microglia. Another stimulus affecting the protein composition of microglial EVs is Wnt3a. Wnt3a stimulation in fact enriched EVs with proteins associated with cellular metabolism, cell architecture and protein synthesis [47], as well as ubiquitin and proteasome components (proteasome subunit B type 7 and type 2), suggesting a role of microglia EVs in the clearance of the extracellular space [47].

A summary of the factors modulating microglial EVs is reported in Table 1.

In addition to metabolic enzymes, chaperones and tetraspanins, microglia-derived EVs also carry membrane receptors and specific cellular markers such as CD13 and MCT-1 [48]. The lipid composition of EVs changes with their dimensions and affects their content. The most abundant lipids present in EVs are cholesterol (CHOL), sphingomyelin (SM), glycol sphingolipids and phosphatidylserine (PS). sEVs are enriched in CHOL and SM [49]. Even if there is no precise evidence for a specific functional role of lipids in microglia-derived EVs, one study demonstrates that the lipid fraction of microglial EVs affects the maturation of oligodendrocyte precursor cells (OPCs) [50]. In particular, sphingosine-1-phosphate (S1P) acts as an attractive guide for OPC migration to myelin lesion sites [50]. Furthermore, lipids in microglia-derived EVs recognize and target different recipient cells [47]. Another lipid enriched in microglia-derived EVs is N-arachidonoylethanolamine (N-AEA), which modulates synaptic transmission. N-AEA is enriched in EVs derived both from N9 cells and primary rat microglia [51].

Different studies highlight the role of EVs in miRNA transfer in the brain [52,53]. The miRNA content in microglia-derived EVs revealed a two-fold enrichment of miR-1860, miR-1705, miR-2284y-6, miR-146a, miR-858, and miR-7718 [54]. Different miRNAs are present in EVs according to different activation states of microglial cells [52]. For example, miR-146-5p, miR-181a and miR-223 are increased upon LPS treatment [52]. These miRNAs have synaptic targets: for example, miR-146 downregulates synaptotagmin 1 (Syt1) and neuroligin 1 (Nlg1) levels in neurons, with a reduction of dendritic spines and synaptic density [52]. In addition, EVs derived from IL-4-stimulated BV2 cells are enriched in miR-26a that promotes angiogenesis [55], revealing another microglial EV-regulated cellular function.

Microglial-derived EVs contribute to maintaining microglia homeostasis influencing their phenotype and gene expression profile. Several studies were conducted to elucidate specific functions regulated by EVs derived from microglial cells under different states of activation, as stated in the previous section. Regardless of the status of the donor cells, microglial EVs modulate, in the target cells, the expression of crucial genes that regulate molecular intracellular pathways such as neuroinflammation (by activating the multimers of the inflammasome), apoptosis and autophagy [56].

As one example, during cellular stress, microglial EVs, acting on other microglial cells, increase the expression of the microtubule-associated protein 1A/1B light chain 3B isoform II (LC3B-II), which is an autophagic marker [56].

Microglia-derived EVs were also investigated for their contribution to neuroprotection. Neuroprotection is defined as the ability to avoid neuronal cell death by inhibiting the signaling pathways activated by cell dysfunction and cell death under pathological conditions. Microglia-derived EVs affect neurite outgrowth in rat primary neurons, due to the transfer of six different miRNAs that are able to regulate the physiological regenerative process in neurons [56].

## 4. Microglial EVs in Brain Tumors

In GBM, microglia and infiltrating macrophages represent about 30% of total cells in the tumor mass, contributing to the early anti-tumor immune response and later playing a role in supporting cancer growth [57,58]. In fact, brain tumor cells attract microglia by secreting factors such as cytokines, growth factors, chemokines and colony-stimulating factors [58], which induce phenotypical and genetical switch of microglia toward a pro-tumorigenic ally. Microglia and infiltrating macrophages, modified by cancer cells, are defined as tumor-associated myeloid cells (TAMCs) [59].

In addition to secreted molecules, microglia-tumor communication is also mediated by gap junctions, tunneling nanotubes and EVs [60]. EVs make possible communication along distant sites and, importantly, also permit bidirectional cell-to-cell communication [60]. As stated before, EVs can deliver not only soluble proteins but also a wide variety of coding and non-coding RNAs that can alter the gene expression of the target cells [60].

## 5. EV-Mediated Communication between GBM and the Brain Tumor Environment

GBM is the most aggressive brain tumor and represents an important research and medical challenge [1]. EVs released by GBM cells have specific cargos that favor tumor propagation. They contain oncogenes, such as the epidermal growth factor receptor (EGFRvIII), that induce the expression of other oncogenes (i.e., p27 and Bcl-xL) and the activation of pro-tumoral pathways (i.e., AKT and ERK1/2 phosphorylation) [61]. Another mechanism of EV-mediated progression of glioma is the transfer of the RNA-binding motif 11 (RBM11) [62], a pro-tumoral protein that promotes invasion and proliferation [63] by apoptotic glioma cells. Additionally, EVs derived from glioma stem cells (GSCs) contribute to maintaining the GBM cellular heterogeneity (a peculiarity of high malignancy) [64]. In addition to the horizontal propagation of the oncogenic activity, glioma-released EVs also convey materials to non-tumoral cells [65], such as astrocytes, neurons, and endothelial and immune cells.

Astrocytes stimulated by GBM-derived EVs acquire a tumor-supportive phenotype [6]. In addition, GBM-derived EVs enhance the proliferation and the transformation of astrocytes by RNAs able to reprogram metabolic activity [66]. Conversely, exosome-mediated transport of miR-19a from astrocytes to tumor cells critically results in PTEN downregulation and tumor growth [67,68].

GBM-derived EVs can regulate neuronal excitability [69] and miRs transfer (miR-148a and miR-9-5p) promotes angiogenesis in endothelial cells [70,71,72,73]. In bidirectional communication, brain endothelial cells release EVs containing tetraspanin CD9 to GBM [74], enhancing tumor progression through the inhibition of ubiquitination of IL6 receptor gp30 and promoting the activation of the signal transducer and activator of transcription 3 (STAT3) [75,76,77,78]. Endothelial-derived EVs overexpressing the tumor suppressor esophageal cancer-related gene-4 (ECRG4) inhibit glioma cell proliferation [79].

GBM-derived EVs suppress the activation of T lymphocytes, partially through the programmed death ligand-1 (PD-L1) [80], a key player of the tumor immune escape in many cancers including GBM [81,82]. Macrophages stimulated with GBM-derived EVs acquire a pro-tumoral phenotype (i.e., overexpression of Arg-1 and IL-10 and downregulation of iNOS and TNF-α), in part through miR-10b-5p [83]. Furthermore, GBM-released EVs also block the clonal proliferation of T cells by transferring CD73 [84], which inhibits aerobic glycolysis [85].

GBM-derived EVs are involved in the initiation of the tumor-supportive TAMC phenotype [86]. TAMCs efficiently engulf EVs, as visualized in vivo, in glioma-bearing mice [87]. GBM-derived EVs increase the phagocytic activity of microglia towards the extracellular matrix, creating free space in the brain parenchyma [88]. One of the mechanisms involved is the transfer of the membrane type 1-matrix metalloproteinase (MT1-MMP) [88], which degrades the proteins of the extracellular matrix. GBM-derived EVs affect TAMC proliferation both in vivo and in vitro [89]. GBM-derived EVs contain miR-21 [90] which downregulates the target tumor-suppressive gene Btg2 (B cell translocation gene 2) [91] in microglial cells. Moreover, they transfer the Wilms tumor-1 (WT1) protein to microglial cells, upregulating the expression of thrombospondin-1 (Thbs-1), a key promoter of angiogenesis [92]. GSCs represent a cell subpopulation relevant for the tumor resistance to radiations and release EVs able to polarize microglial cells towards a pro-tumoral phenotype, specifically by transferring miR-504 that inhibits the putative onco-suppressor gene grb10 (growth factor receptor-bound protein 10) [90]. miR-504 also increases the pro-tumoral markers CD209 and TGF-β and decreases the expression of the anti-tumoral marker genes CD86 and TNF-α [90]. GBM-released EVs reprogram microglia towards a pro-tumoral phenotype also through the long noncoding RNA (lncRNA) associated with temozolomide (TMZ) (lnc-TALC). lnc-TALC competes with miR-20b-3p by inducing the expression of Stat3, which, in turn, activates the expression of the DNA repair enzyme O^6^-methylguanine-DNA methyltransferase (MGMT) [93]. lnc-TALC also increases Arg-1, CD163, TGFβ, IL4, and IL10 [94], enhancing the pro-tumoral phenotype of microglia.

On the other side, several pieces of evidence demonstrate that microglial-released EVs can affect the TME and the effects induced by microglial EVs vary with the activation state of these cells [95]. Specifically, m/lEVs released by LPS/INFγ-activated microglia contain transcripts for several inflammation-related genes that reduce the pro-tumoral efficacy of TAMCs. In contrast, m/lEVs released by IL4-stimulated microglia enhance the pro-tumoral phenotype of TAMCs [95]. We mentioned before that the biogenesis process and the cargo differ in sEVs and m/lEVs. In fact, in contrast with m/lEVs, sEVs released by both LPS/INFγ and IL4-stimulated microglial cells exert anti-tumoral effects in a mouse model of glioma, reducing tumor mass and prolonging mice survival [96]. Similarly, sEVs released by unstimulated microglial cells reduce the invasion of glioma cells in 3D-spheroid cultures [97]. The anti-glioma effect of microglial-derived sEVs can be mediated by miR124. This specific miR enhances the expression of the glutamate transporter GLT1 on astrocytes, with increased clearance of glutamate in the synaptic cleft. Glioma cells are in fact able to release neurotoxic amounts of glutamate that promote neuronal death and permit the invasion of tumor cells [96]. 

EV-mediated delivery of miRNAs could represent a promising approach to contrast the growth of brain tumors. In a 3D-microfluidic GBM microenvironment, miR-124-loaded EVs reduce tumor growth and inhibit microglia polarization towards a pro-tumoral phenotype [98]. Furthermore, microglia EVs can be engineered to deliver drugs to glioma cells, taking advantage of their ability to target the brain tumor mass [99]. Specifically, microglia cells were engineered to release EVs enriched in paclitaxel (PTX), a pro-apoptotic compound that blocks the cell cycle at the G2/M phase [100]. Loading PTX in EVs overcomes its low BBB permeability [101] and could be used for other drugs with a reduced capability to enter brain parenchyma.

A summary of pro-tumorigenic and anti-tumorigenic factors transported via EVs in glioma is reported in Table 2.

## 6. Conclusions 

Microglial-released EVs are active players in maintaining or restoring CNS homeostasis (Figure 1). They can be heterogeneous depending on the state of activation of microglial donor cells and, in turn, depending on the microenvironment stimuli. In addition, EVs have stable structures and high stability in the extracellular space. All these features make microglial EVs potential tools to deliver therapeutics in the CNS with the purpose of interfering with the growth and spread of tumors in the brain parenchyma. 

However, the current knowledge on the effects of EVs in the context of brain tumors is not sufficient to propose their immediate use in clinics and the data summarized in this review suggest that a more detailed description of the content of EVs in the different experimental conditions and in patients, and a better understanding of how the formation process affects their cargo, is necessary to move forward in EV-based therapeutic applications. In particular, engineered EVs could enhance the efficacy of EV-mediated anti-tumorigenic pathways. Finally, we conclude that the knowledge of central and peripheral EV content in the glioma landscape could contribute to an early diagnosis and, possibly, to future personalized therapies.

## Figures and Tables

**Figure 1 ijms-23-13165-f001:**
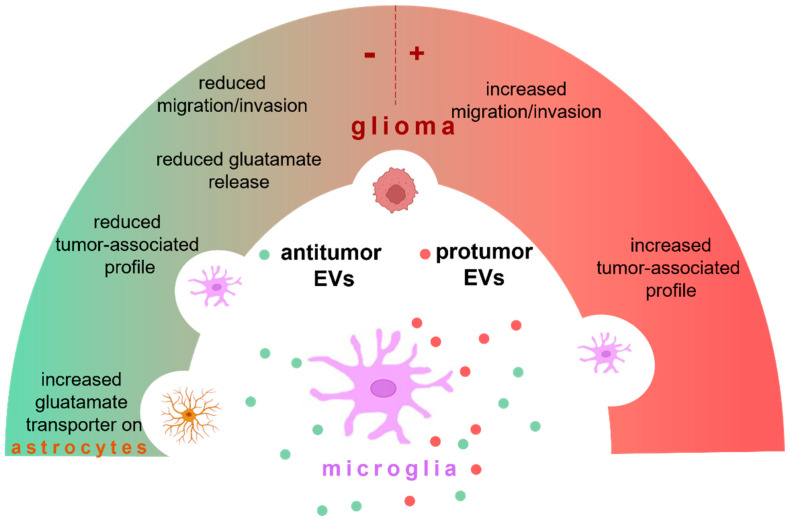
Microglia-derived EVs in glioma.

**Table 1 ijms-23-13165-t001:** Modulating factors, specific receptors, and effects on microglial EVs.

Factor	Receptor	Effect on Microglial EVs	Reference
ATP	P2X7	increased production	[39]
		enrichment in proteins regulating cell adhesion, phagocytosis, energy metabolism, autophagy	[39]
		enrichment in Tau protein	[40]
		enrichment in C1q	[41]
capsaicin	TRPVI	increased production	[42]
α-syn		increased production	[43]
		enrichment in TNF-α and (MCH)-II receptor	[43]
LPS		enrichment in pro-inflammatory chemokines, cytokines and miRs	[45]
		enrichment in proteins regulating chemotaxis	[45]
		enrichment in proteins activating the NFkB pathway	[45]
		enrichment in proteins regulating the metabolism	[45]
5-HT	5-HTR2, 5HTR4	increased production	[46]
Wnt3		enrichment in proteins regulating cellular metabolism, cell architecture and protein synthesis	[47]
		enrichment in ubiquitin, proteasome subunit B type 7 and type 2	[47]

**Table 2 ijms-23-13165-t002:** Pro- and anti-tumorigenic factors transported by EVs in glioma microenvironment. Donor cells are listed in columns and target cells in rows.

**PRO-** **TUMOR EVs**	**GBM**	**Microglia/** **Macrophages**	**Astrocytes**	**Endothelial Cells**	**T Cells**
**GBM**	EGFRvIII [61]RBM11 [62]	miR-10b-5p [83]MT1-MMP [88]miR-21 [90]WT1 [92]miR-504 [90]lnc-TALC [93,94]	RNAs for metabolic activity [66]	miR-148a [70,71,72,73] miR-9-5p [70,71,72,73]	PD-L1 [80] CD73 [84]
**Astrocytes**	miR-19a [67,68]				
**Endothelial cells**	CD9 [74,75,76,77,78]				
**ANTI-** **TUMOR EVs**	**GBM**	**Microglia/** **macrophages**	**Astrocyte**		
**Endothelial cells**	ECRG4 [79]				
**Microglia/macrophages**	miR124 [96,98]	inflammation-related genes [95]miR124 [98]	miR124 [96]

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
