# Peer review of "Microglial Extracellular Vesicles as Modulators of Brain Microenvironment in Glioma"

_ijms, 2022, doi:10.3390/ijms232113165_

Round 1

Reviewer 1 Report (New Reviewer)

Overall, the review was informative and well-organized. Clear focus of microglia-derived EVs and relationship to tumor environment.

However, many paragraphs contained so many different molecules and abbreviations that it is difficult to keep track and see the big picture. It is recommended that  these pathways be listed in table by showing anti- vs pro-tumorigenic or in a signaling pathway figure to illustrate the points.

Conclusions could be expanded upon with more specific recommendations for future investigations.

Line 230: ref 58, close parenthesis

Abbreviation list is not complete.

Author Response

Reviewer 2 Report (New Reviewer)

This review summarized the relationship between the microglial extracellular vesicles and glioma. The author described EVs as active players in maintaining or restoring CNS homeostasis, the contribution of EVs released by microglia to the growth and maintenance of GBM, and to antitumor effects. The review is well demonstrated and only a few minor points need to be addressed before publication.

Here are the comments:

1. In Section 1, Line 56-63. Please adjust the figure caption's font size and make it closer to the figures as it looks like the main text in its present form. 

Also, statistics specifically about glioma - incidence, survival rates, etc - would be a useful addition here.

2. In Section 2, from Line 97. It would be better to separate them into two subsections 2.1 and 2.2 to discuss the mlEVs and sEVs, and they should be expanded a little bit

3. In Section 4, Line 232. It will be a little bit confusing using TAMs as the abortion of tumor-associated myeloid cells. Usually, TAMs will refer to tumor-associated macrophages and TAMCs refer to tumor-associated myeloid cells. Although I can see the authors have clearly defined the abbreviations, however, if the readers search for TAMs they might more likely expect to see tumor-associated macrophages.

4. In Section 5, Line 271, there is a dash at the beginning of this paragraph.

Author Response

Reviewer 3 Report (New Reviewer)

Dear Editor,

Thank you very much for providing the opportunity for reviewing this manuscript. This review paper summarized the function of microglial-derived EVs as modulators for GBM. The manuscript is nicely organized and well written. It covers the main findings in the field.

The only concern that I have is that whether the authors can separate the molecular identities and functions of miroglial-derived sEVs and m/lEVs in GBM since they have distinct biogenesis pathways and likely present heterogeneity in molecular contents. It will be better to have a summary figure or table to make a comparison of biophysical properties, molecular contents, and functions of these two types miroglial-derived EVs.  

Author Response

Reviewer 4 Report (New Reviewer)

This is a review of the bidirectional communication between malignant gliomas, esp glioblastoma multiforme (GBM), and microglial cells (resident immune cells) in the brain, mediated by formation, release and uptake of extracellular vesicles (EV's) by both cell types. The authors provide an easily read state-of-the-art review of EV's- how they are formed and what cargoes they can transport to other cells. As the authors discuss, EV's from microglia in a GBM environment contribute to reduction in immune surveillance of the tumor; and, the GBM tumor cells alter the microglia in the GBM environment so as to facilitate the survival and spread of the GBM. We are clearly at the beginning of understanding this complex, two-way "mutual admiration society" of brain microglia and GBM tumor. The authors provide the insight that this knowledge will likely lead to better treatment of GBM, an otherwise poorly treated and lethal brain neoplasm.

The paper is thus timely and should stimulate further investigations (two characteristics of good review articles), There are a very few and minor grammatical errors that need correcting, but these should be easily identified by careful editing. I hope the authors decide to update this review in 2-3 years.

Author Response

This manuscript is a resubmission of an earlier submission. The following is a list of the peer review reports and author responses from that submission.

Round 1

Reviewer 1 Report

The review is very interesting especially for the therapeutic indications it takes into consideration. However, in my opinion, there is a paragraph missing in which we try to identify the specific characteristics of microglial exosomal vesicles.

In other words, it would be necessary to deepen the concept of the importance of identifying EV markers specific to a specific cell type, as in this case of miscroglia. Therefore, I advise the authors to enrich the manuscript according to these indications.

Finally, I recommend inserting, at the end of the review, a paragraph (in alphabetical order) with all the abbreviations used in the text.

Below are indications of minor importance that need to be changed in the text.

Line 32 CNS is used for the first time, please insert Central Nervous System.

Line 103-107: please insert the appropriate references.

Line 140: Chande “the” with “The”.

Lines 285, 286, 289: please enter the corresponding words to the acronyms

Author Response

Please, find the response in the attached file.

Reviewer 2 Report

The title of this review deals with Microglial extracellular vesicles as modulators of brain microenvironment in glioma. In the very short Abstract, though, the cancer deals with one line out of eight; in the text a similar ratio occurs. As long as a review is so poorly organised and presented it cannot be published in a prestigious Journal. 

Author Response

(The authors gave the same response as above.)
